# The Safety Profiles of Two First-Generation NTRK Inhibitors: Analysis of Individual Case Safety Reports from the FDA Adverse Event Reporting System (FAERS) Database

**DOI:** 10.3390/biomedicines11092538

**Published:** 2023-09-15

**Authors:** Valerio Liguori, Mario Gaio, Alessia Zinzi, Cecilia Cagnotta, Consiglia Riccardi, Giovanni Docimo, Annalisa Capuano

**Affiliations:** 1Campania Regional Centre for Pharmacovigilance and Pharmacoepidemiology, 80138 Naples, Italy; valerio.liguori@unicampania.it (V.L.); alessia.zinzi@unicampania.it (A.Z.); ceciliacagnottacg@gmail.com (C.C.); consiglia.riccardi@unicampania.it (C.R.); annalisa.capuano@unicampania.it (A.C.); 2Section of Pharmacology “L. Donatelli”, Department of Experimental Medicine, University of Campania “Luigi Vanvitelli”, 80138 Naples, Italy; 3Department of Advanced Medical and Surgical Sciences, University of Campania “L. Vanvitelli”, 80138 Naples, Italy; giovanni.docimo@unicampania.it

**Keywords:** TRK, NTRK, agnostic drugs, precision medicine, larotrectinib, entrectinib

## Abstract

The first-generation tropomyosin receptor kinase (TRK) inhibitors, larotrectinib and entrectinib, represent exciting new developments in cancer treatment that offer relevant, rapid, and long-lasting clinical benefits. Larotrectinib and entrectinib are recommended as first-line treatments for locally advanced or metastatic non-small cell lung cancer (NSCLC) patients with positive TRK gene fusions. In this study, using the U.S. Food and Drug Administration (FDA) Adverse Event Reporting System (FAERS) database between 2019 and 2022, a retrospective analysis was conducted to evaluate the safety profiles of these drugs. During our study period, 807 individual case safety reports (ICSRs) related to larotrectinib or entrectinib were retrieved from the FAERS database, of which 48.7% referred to females and 24.7% referred to adult patients (18–64 years) with a median age of 61.0 years. A total of 1728 adverse drug reactions (ADRs) were identified. The most frequently reported ADRs were dizziness and pain, which belong to the System Organ Classes (SOCs) “nervous system disorders” and “general disorders and administration site conditions”. Regarding all ADRs, the median time to onset was 37.0 days for larotrectinib and 12.0 days for entrectinib. No evident safety concerns emerged in the long-term safety profiles (>365 days). Only 18 ICSRs were related to pediatric populations (≤16 years), of which 94.0% of the ICSRs were related to larotrectinib. The median age was 10.5 years, while most patients were female (44.4%). Our results show favorable risk-benefit profiles for larotrectinib and entrectinib. Considering the increased use of neurotrophic tyrosine receptor kinase (NTRK) inhibitors, continuous safety monitoring of larotrectinib and entrectinib is required for the detection of possible new adverse drug reactions.

## 1. Introduction

For a long time, the Food and Drug Administration (FDA) has evaluated and authorized medications to treat various types of cancer based on where the cancer has developed in the body. In recent years, the advent of precision medicine has globally contributed to the development of alternative drugs for treatments that are aimed towards specific targets [1,2]. In this regard, precision oncology has exploited next-generation sequencing (NGS) to be able to design alternative molecules defined as tumor-agnostic drugs or tissue-agnostic drugs [3]. The specificity of these molecules and their mechanisms of action lies in the fact that they are active against different forms of oncogenic-dependent tumors, regardless of where the cancer begins in the human body [4,5]. Therefore, the advantage of these new drugs is the ability to treat any type of cancer considering that the drugs are highly selective for specific molecular alterations [4]. On 23 May 2017, the FDA approved pembrolizumab, a monoclonal antibody (mAb) targeting programmed cell death 1 (PD-1) receptor, as the first tumor-agnostic treatment [6]. Afterwards, two first-generation tropomyosin receptor kinase (TRK) inhibitors received tumor-agnostic approvals from the FDA, i.e., larotrectinib and entrectinib [7,8]. Larotrectinib is a highly selective and central nervous system (CNS)-active TRK inhibitor, while entrectinib is a potent small-molecule, selective inhibitor of TRKA/B/C and ROS1 (ROS proto-oncogene 1, receptor tyrosine kinase) (Figure 1) [9,10].

Both larotrectinib and entrectinib are indicated for the treatment of advanced or metastatic solid tumors harboring neurotrophic TRK (NTRK) gene fusions in adults and pediatric patients (Table 1) [7,8,11]. This recommendation has been based on the positive results of studies that enrolled patients affected by soft tissue sarcomas, lung cancer, and salivary gland cancer [12]. Larotrectinib received accelerated approval by the FDA on 26 November 2018, based on the results of a combined analysis of three clinical trials (LOXO-TRK-14001, SCOUT, and NAVIGATE) published by Drilon and colleagues [10,13,14,15]. Entrectinib was approved by the FDA on 15 August 2019, after a pooled analysis of three single-arm studies (ALKA-372-001, STARTRK-1 e STARTRK-2) published by Doebele et al. [10,16,17]. The Committee for Medicinal Products for Human Use (CHMP) of the European Medical Agency (EMA) also recommended approvals for larotrectinib and entrectinib on 19 September 2019 and on 31 July 2020, respectively [7,8].

Despite their clinical efficacy, larotrectinib and entrectinib can induce even potentially serious adverse reactions. Therefore, risk management plans (RMPs) have been presented by the marketing holders to ensure rigorous effectiveness and safety monitoring of larotrectinib and entrectinib in real-world settings [7,8]. Regarding the safety profiles of these drugs, treatment-related adverse events (TRAEs) have mainly been Grade 1 or 2. One of the most frequent safety concerns is severe neurologic reactions, as already reported in the Summary of Product Characteristics (SmPC) and in the Medicine European Public Assessment Report (EPAR) of larotrectinib and entrectinib [7,8]. However, evidence about other types of serious adverse events and about the pediatric population are still scant. Finally, tumors generated by NTRK fusions, similar to many other metastatic tumors, can relapse leading to acquired resistance and necessitating the selection of new targeted therapy. It is hopeful that second-generation TRK inhibitors such as selitrectinib and repotrectinib are being tested in clinical trials to overcome these recurrent resistance mutations [18]. Given that larotrectinib and entrectinib will become more frequently used in daily clinical practice, this pharmacovigilance study aims to provide an analysis of safety reports collected in the FDA Adverse Event Reporting System (FAERS) database. This study also aims to assess the long-term safety profiles (>365 days) of larotrectinib and entrectinib and their use in the pediatric population (≤16 years).

## 2. Materials and Methods

### 2.1. Data Source

Data on individual case safety reports (ICSRs) with larotrectinib or entrectinib as the suspect drugs were retrieved from the FDA Adverse Event Reporting System (FAERS) database. The FAERS is a passive surveillance system for collecting worldwide reports of suspected adverse drug reactions (ADRs) submitted by healthcare professionals, consumers, and pharmaceutical manufacturers [19]. The FDA publishes FAERS files every quarter (i.e., four files each year (Q1, Q2, Q3, and Q4)) which are available to the public in different ways, i.e., a user-friendly public dashboard or raw quarterly data downloadable as ASCII or XML files. All the ASCII quarterly data files from the FAERS database were downloaded from the first quarter of 2019 (2019Q1) to the fourth quarter of 2022 (2022Q4). Each quarterly data file includes the following seven datasets: DEMO (patient demographic and administrative information), DRUG (drug information for all medications reported for the event), OUTC (patient outcomes for the event), RPSR (report sources for the event), THER (drug therapy start dates and end dates for the reported suspect drugs), INDI (indication information), and REAC (both the INDI and REAC datasets contain the Medical Dictionary for Regulatory Activities (MedDRA) terminology). Suspected ADRs are recorded as preferred terms (PTs) according to the MedDRA terminology [20]. The MedDRA is a standardized medical terminology that is used globally for the classification of adverse event information associated with medical products, including pharmaceuticals, biologics, vaccines, and medical devices. It provides a structured and comprehensive language for accurate recording and analysis of adverse events in clinical trials, post-marketing surveillance, and regulatory submissions. The MedDRA’s hierarchical structure consists of multiple levels—from the Low-Level Terms level to the System Organ Class level—with each level providing progressively broader categories for classifying medical terminology. Each PT is associated with a High-Level Term (HLT), a High-Level Group Term (HLGT), and a System Organ Class (SOC) level in the MedDRA.

A validated software (R, version 4.3.0, R Development Core Team) was used to select and merge the data. Specifically, first, we selected all individual case safety reports including larotrectinib or entrectinib as the suspect drug. Then, we linked data by using the primary identifier of each ICSR as the key to linkage. The obtained data were manually scrutinized to remove duplicates caused by concurrent reporting. We identified the duplicates by using the following variables: age, sex, starting and ending date of the treatment, the indication, the date of onset, and looking at the similarities between the ADRs.

### 2.2. Descriptive Analysis

In the present post-marketing safety surveillance survey, we analyzed all ICSRs with larotrectinib or entrectinib as the suspect drugs in the period between 1 January 2019 and 31 December 2022. We performed a descriptive analysis providing information about ICSRs (reporting year, reporter type, and the country for regulatory purposes), baseline demographic characteristics of patients (age and sex), suspect drugs (indication of use), and ADRs (number, time to onset, and frequencies by SOC).

A subgroup analysis of pediatric patients (≤16 years) was performed. Moreover, to identify ICSRs reporting any neurological ADRs in pediatric patients in accordance with the larotrectinib’s and entrectinib’s risk management plans, we used the following coding system: “cognitive and attention disorders and disturbances” (HLGT), “developmental motor skills disorders” (HLT), “developmental disorders cognitive” (HLT), “memory loss (excluding dementia)” (HLT), “mental impairment (excluding dementia and memory loss)” (HLT). The outcomes of ADRs were classified as “death”, “life-threatening”, “hospitalization” (initial or prolonged), “disability”, “congenital anomaly”, “required intervention to prevent permanent impairment/damage”, and “other serious” (important medical event).

Lastly, a subanalysis of the ICSRs including long-term safety information (time to onset of ADRs >365 days) was performed.

## 3. Results

The clinical and demographic characteristics of the ICSRs collected from the FAERS database are reported in Table 2. From 1 January 2019 to 31 December 2022, 807 ICSRs related to larotrectinib (*N* = 405) and entrectinib (*N* = 402) were retrieved from the FAERS database, of which 393 (48.7%) ICSRs referred to females and 24.7% of the ICSRs referred to adult patients (18–64 years) with a median age of 61.0 years (48.7–70.0 IQR). Moreover, the majority of the ICSRs were issued by consumers (*N* = 289, 35.8%), followed by physicians (*N* = 268, 33.2%), and health professionals (HCP) (*N* = 145, 18.0%). The United States was the main reporting country, and 2022 was the highest reporting year.

Looking at the indications of use as reported in the analyzed ICSRs, larotrectinib and entrectinib were used mostly for the treatment of respiratory tract and pleural neoplasms malignant cell type unspecified NEC (not elsewhere classified) (*N* = 154, 19.1%), followed by non-small cell neoplasms malignant of the respiratory tract cell type specified (*N* = 124, 15.4%), therapeutic procedures NEC (*N* = 74, 9.2%), and neoplasms malignant site unspecified NEC (*N* = 42, 5.2%). In 114 ICSRs (14.1%) the indication of use was not reported (Table 3).

We analyzed a total of 1728 reported adverse events. The apparent numerical discrepancy between ICSRs and adverse events is simply due to the possibility of reporting more than one event per ICSR. The most reported ADRs were included in the SOCs “general disorders and administration site conditions” (larotrectinib *N* = 148 (17.8%) and entrectinib *N* = 154 (17.2%)) and “nervous system disorders” (larotrectinib *N* = 103 (12.4%) and entrectinib *N* = 118 (13.2%)). On the one hand, the most reported PTs related to larotrectinib among the SOC “nervous system disorders” were dizziness (*N* = 30, 3.6%) and neuropathy peripheral (*N* = 21, 2.5%), and the most reported PTs among the SOC “general disorders and administration site conditions” were pain (*N* = 29, 3.5%) and fatigue (*N* = 26, 3.1%). On the other hand, the most reported PTs for entrectinib among the SOC “general disorders and administration site conditions” were death (*N* = 55, 35.7%) and disease progression (*N* = 19, 12.3%), while the most reported PTs among the SOC “nervous system disorders” were dizziness (*N* = 42, 35.6%) and taste disorder (*N* = 12, 10.2%) (Figure 2).

Our analysis also focused on the long-term safety profiles. Specifically, we analyzed all the suspected adverse events that occurred after 365 days of the larotrectinib or entrectinib administration. The median times to onset of ADRs related to larotrectinib and entrectinib were 37 days (15.5–92.0) and 12 days (3.7–40.3), respectively (Figure 3). Only 11 ICSRs included PTs with a time to onset (TTO) >365 days.

The most reported SOCs after 365 days since the beginning of treatments with larotrectinib or entrectinib were “general disorders and administration site conditions” (*N* = 8, 22.8%), followed by “nervous system disorders” (*N* = 7, 20.0%) and “musculoskeletal and connective tissue disorders” (*N* = 6, 17.1%). In more detail, dizziness, paraesthesia, myalgia, fatigue, asthenia, muscle spasm, and muscle tightness were the PTs most reported for each above-mentioned SOC (Figure 4 and Table 4).

A subgroup analysis was performed for pediatric ICSRs (*N* = 18 of 807 ICSRs). During the study period, *N* = 17 ICSRs with larotrectinib and *N* = 1 ICSR with entrectinib as the suspect drugs were retrieved from the FAERS database for a total of 37 ADRs. The median age of patients who experienced ADRs was 10.5 years (IQR 0.87–15.25) and 44.4% of the patients were female (in six cases the sex was not reported). The most reported indication of use of larotrectinib was sarcoma, while in the only case associated with the use of entrectinib, the indication of use was a neuroendocrine tumor. Furthermore, the majority ICSRs described the occurrence of suspected ADRs belonging to the SOCs “gastrointestinal disorders” (e.g., vomiting and nausea), “general disorders and administration site conditions” (e.g., pyrexia), “investigations” (e.g., weight increased), “blood and lymphatic system disorders” (e.g., blood disorder), and “neoplasms benign, malignant and unspecified (including cysts and polyps)” (e.g., fibrosarcoma metastatic). A total of 55.5% of all pediatric ICSRs (*N* = 10) was reported as serious and almost half of them were included in the “other medically important condition” seriousness criteria (Table 5). Only two ICSRs reported TTO data. In the first case, ADRs occurred after 4 days, while in the second case, ADRs occurred after 5 days. No ICSRs reporting neurological ADRs in pediatric patients were reported.

## 4. Discussion

In this study, we analyzed all ICSRs related to larotrectinib and entrectinib through an analysis of data from the FAERS database, a U.S. national drug safety passive monitoring system database. In the present analysis, the majority of ADRs were mainly related to female and adult patients for which the consumer was the principal source. In line with our results, some studies on ADRs spontaneously reported in pharmacovigilance databases have suggested that, independently by drug classes, women are more susceptible to experiencing ADRs [21,22,23]. This aspect could be explained by increased drug use in women compared with men, increased polypharmacy and consequently drug–drug interactions, or actual sex differences in pharmacokinetics or pharmacodynamics that make women more susceptible to ADRs compared with men [24,25,26]. From 2019 until 2022, there has been a constant increase in ICSR reporting, probably related to the increased utilization of both NTRK inhibitors. In this study, the main therapeutic indications were related to lung cancer. These findings are not surprising given that, as mentioned in the SmPC, larotrectinib and entrectinib are primarily indicated for the treatment of solid tumors, with entrectinib especially authorized for the treatment of NSCLC [7,8]. This condition is more common in female adult patients. According to data given by the American Cancer Society Trusted Source in 2023, the majority of patients diagnosed with lung cancer are 65 and older, and lung cancer affects more females (120,790 new cases) than males (117,550 new cases) [27]. In our analysis, the most common neurological adverse events were dizziness and peripheral neuropathy. TRAEs of dizziness as well as peripheral neuropathy have been frequently observed in clinical trials of larotrectinib and entrectinib [16,28]. Regarding the ICSRs involving entrectinib, our data showed that “death” and “disease progression” were the most often reported PTs among the SOC “general disorders and administration site conditions”. According to the MedDRA guide, death terms may have additional secondary links to the related site or etiology SOCs [29]. Specifically, this SOC represents a class of disorders that encompasses conditions of a general type that result from a disease, the treatment of a disease, or the administration of treatment at a particular site, and are manifested by a characteristic set of symptoms and signs. Therefore, considering the inherent limitations of the FAERS, we cannot derive information on the real cause of death such as the confounding effects of concomitant diseases. The literature data have reported that the observed cases of death after the administration of larotrectinib or entrectinib are usually deemed unrelated to treatment and are often due to acute respiratory failure, cardiorespiratory arrest, or pneumonia [16]. In addition, a crucial point to remember about suspected adverse drug reactions collected in a spontaneous reporting system, especially those that are lethal, is that we cannot be certain of their causal association with medicine; indeed, adverse events occurring during pharmacological therapy are not necessarily related to it [30]. According to the literature, patients with advanced cancer are frequently not treated or are treated late [31]. Regarding larotrectinib and entrectinib, it is important to highlight that these drugs are often used in patients who have a disease that is locally advanced or metastatic or who have no satisfactory treatment options. Therefore, considering the above premises, we cannot rule out a direct role of underlying disease (often in an advanced stage) in the occurrence of death. In support of this hypothesis, there are many clinical settings in which the local relapse biologically triggers cancer progression and consequentially death, and therefore, progression disease does not necessarily indicate treatment failure [31,32]. According to the EPAR of larotrectinib, no cardiac risks have been identified during clinical trials. On the contrary, for entrectinib, cases of congestive heart failure and QT prolongation have been observed [7,8]. Therefore, we also focused on ADRs of special interest, especially those referred to as cardiac ADRs. In our analysis, a few cases of congestive heart failure occurred with a frequency of less than 2% (0.7%) and no cases of QT prolongation were observed. Only one cardiac ADR classified as cardiac disease was identified in the long-term analysis. The median time to onset of adverse events was longer in patients treated with larotrectinib (37.0 days) than in those treated with entrectinib (12.0 days). To date, real-world data on TTO are still scant. Most of the evidence available for both drugs derive from preauthorization studies. For example, as defined in the SmPC of larotrectinib, neurologic adverse events have occurred within the first three months of treatment (range from 0 days to 35.5 months). Regarding entrectinib, TTO is between 0.4 months for ataxia (range from 0.03 months to 28.19 months) and 3.4 months for fracture (range from 0.26 months to 18.5 months). Next, we focused our attention on ICSRs related to larotrectinib and entrectinib in a pediatric population (≤16 years). The data in the literature suggest that entrectinib and larotrectinib are well tolerated in pediatric patients and have shown encouraging antitumor activity in all patients with TRK fusion-positive tumors [33,34]. However, there have been no pharmacovigilance studies on the pediatric population so far, therefore we believe it is critical to assess the tolerability profile of larotrectinib and entrectinib. Almost all pediatric ICSRs indicate larotrectinib as a suspect drug, and our data suggest that the majority of ICSRs have been associated with female patients, confirming that females are more susceptible to develop ADRs during pharmacological treatment [21]. According to our results, the median age of patients treated with larotrectinib or entrectinib reflects the epidemiological data of the disease for which these drugs are used (e.g., fibrosarcoma), showing that these tumors are rare in younger children and become more common with increasing patient age [35]. The majority of ADRs were serious, according to the severity of the existing illness. Consistent with available clinical evidence, our results show a prevalence of vomiting, intestinal obstruction, nausea, and blood disorder which are very common ADRs from larotrectinib administration. Only one entrectinib-related ICSR indicated disease progression as the ADR. This condition is known, given that the risk of progression to more severe disease has been described following therapy with entrectinib, but it does not always associate with treatment [36,37]. To ensure the timely identification of potentially severe adverse events, special attention was given to ADRs that may occur after 365 days from the start of treatment with larotrectinib or entrectinib. As previously reported, even after 365 days, the higher proportion of ADRs was among the SOC “general disorders and administration site conditions” (e.g., fatigue) followed by “nervous system disorders” SOC (e.g., dizziness). Indeed, the only ADRs observed in more than one patient were myalgia and fatigue, which are well described in the SmPC [7,8]. Given the low incidence of all other events identified in our long-term analysis, there was no new safety issue.

## 5. Conclusions

To the best of our knowledge, this is the first report of post-marketing surveillance of the safety profiles of larotrectinib and entrectinib, by using data from the FAERS database, in a real-world setting, focused on adult and pediatric populations and long-term data. The present analysis provides an overview of spontaneous reports of adverse events occurring in adults and pediatric patients treated with larotrectinib or entrectinib, representing a bridge between premarketing data and clinical practice. The results of this three-year period post-marketing surveillance confirm the safety profiles. Considering that both drugs have recently obtained marketing authorization, long-term follow-up studies are strongly needed to evaluate the safety profiles of NTRK inhibitors and, specifically, data from real-life contexts need to be collected.

## 6. Strengths and Limitations

Analyses of large pharmacovigilance databases, such as the FAERS database, allow the extrapolation of important safety information coming from a real-world context [38]. The FAERS database represents a useful and inexpensive tool that provides for better characterization of drug safety profiles and for overcoming intrinsic limits of clinical trials. Indeed, a spontaneous reporting system allows easy identification of specific ADRs that are not detectable during the pre-marketing phase, including rare and serious ADRs. The FAERS has inherent limitations. The spontaneous reporting system is affected by limitations that are mainly related to underreporting and inaccuracy or incompleteness of information [39]. In our opinion, the combination of the first-generation TRK inhibitors’ brief periods of marketing and the relatively limited populations for which they are prescribed may be to blame for the underreporting of cases. First, a causal relationship between drug exposure and an adverse event occurrence cannot be determined, as a causality assessment is not required by spontaneous reporting of adverse drug events to the FAERS. Second, the lack of detailed clinical information on patients (for example, comorbidities, the severity of the underlying illness, and concomitant drugs) limits the ability to control for confounding factors in the ADR occurrence. Specifically, it is well known that both larotrectinib and entrectinib have safety profiles that can be influenced by concomitant medications. For instance, in the case of larotrectinib, P-glycoprotein inhibitors may lead to increased concentrations of larotrectinib in the brain, potentially resulting in an elevated risk of central nervous system-related adverse reactions [7]. In light of the above, this analysis provides preliminary results on the safety profiles of larotrectinib and entrectinib during clinical practice, providing useful information for a risk-benefit profile evaluation.

## Figures and Tables

**Figure 1 biomedicines-11-02538-f001:**
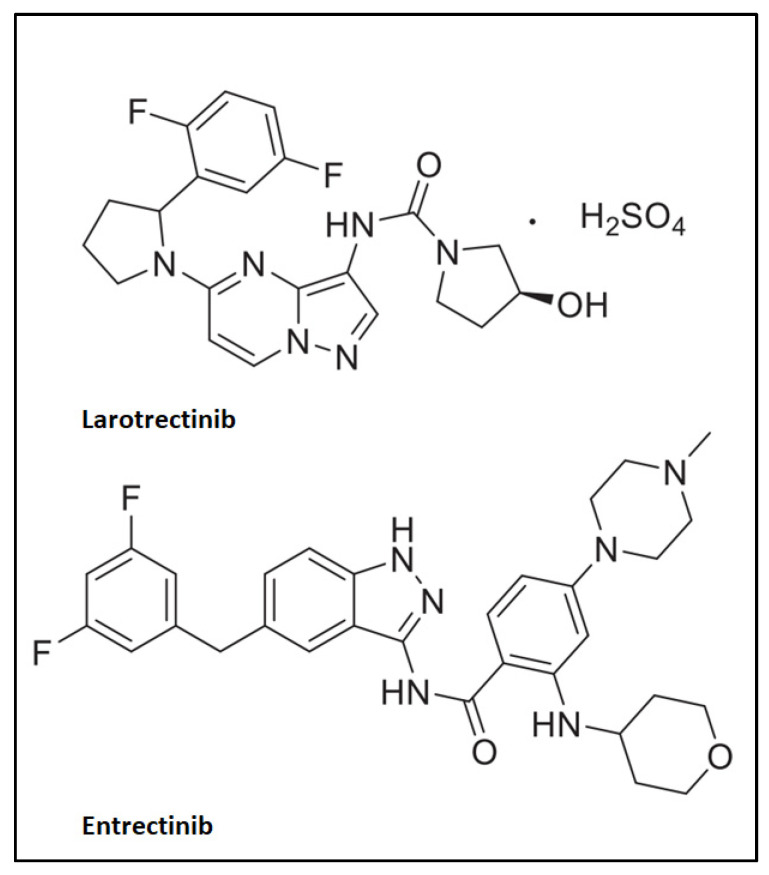
Chemical structures of larotrectinib and entrectinib.

**Figure 2 biomedicines-11-02538-f002:**
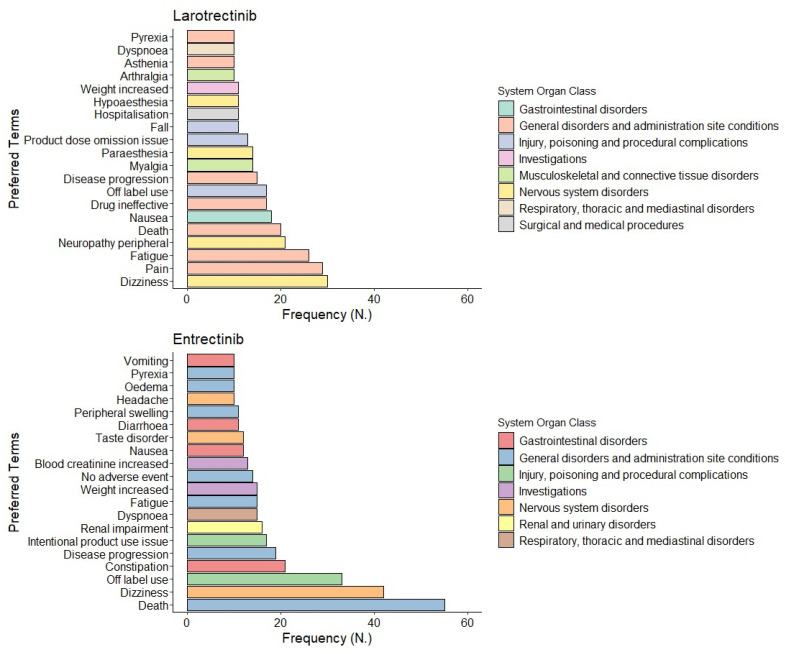
Distribution of the preferred term (PT) (at least accounted for ≥2% of all ADRs) related to each ADR reported in overall individual case safety Reports linked to larotrectinib (*N* = 832) and entrectinib (*N* = 896).

**Figure 3 biomedicines-11-02538-f003:**
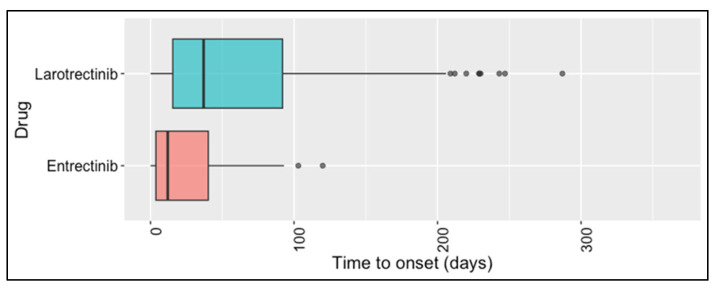
Boxplot of time to onset of ADRs after larotrectinib administration and after entrectinib administration. Time to onset is defined as the time in days from drug administration to the occurrence of an adverse event.

**Figure 4 biomedicines-11-02538-f004:**
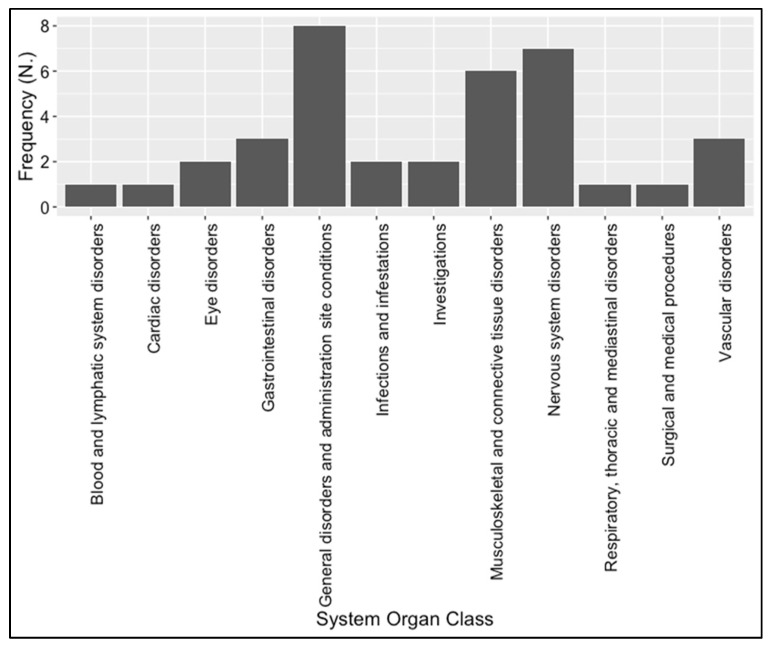
Distribution of the System Organ Class (SOC) related to each ADR reported in overall individual case safety reports linked to larotrectinib and/or entrectinib after 365 days.

**Table 1 biomedicines-11-02538-t001:** Characteristics of NTRK inhibitors.

Agnostic Drug	Date of FDA Approval	Target	Indication(s)	Available as
**Larotrectinib**	26 November 2018	NTRK	Adult and pediatric patients with solid tumors that display a NTRK gene fusion: - who have a disease that is locally advanced, metastatic, or where surgical resection is likely to result in severe morbidity, and - who have no satisfactory treatment options	Hard capsules (100 mg or 25 mg) Oral solution (20 mg/mL)
**Entrectinib**	15 August 2019	NTRK/ROS1	Adult and pediatric patients (≥12 years of age) with solid tumors that display a NTRK gene fusion: - who have a disease that is locally advanced, metastatic, or where surgical resection is likely to result in severe morbidity, and - who have not received a prior NTRK inhibitor - who have no satisfactory treatment options Adult patients with ROS1 -positive, advanced NSCLC not previously treated with ROS1 inhibitors	Hard capsules (200 mg or 100 mg)

NTRK—neurotrophic tropomyosin receptor kinase; ROS1—ROS proto-oncogene 1, receptor tyrosine kinase; NSCLC—non-small cell lung cancer.

**Table 2 biomedicines-11-02538-t002:** Demographic characteristics of individual case safety reports involving larotrectinib or entrectinib recognized in the spontaneous FAERS database from 2019–2022.

		Larotrectinib		Entrectinib		Total	
		N	%	N	%	N	%
**ICSR ^a^**		405	(50.2)	402	(49.8)	807	(100.0)
**Sex**	Female	179	(44.2)	214	(53.2)	393	(48.7)
	Male	159	(39.3)	143	(35.6)	302	(37.4)
	NA	67	(16.5)	45	(11.2)	112	(13.9)
**Age group**	<18	31	(7.7)	3	(0.7%)	34	(4.2)
	18–64	92	(22.7)	107	(26.6)	199	(24.7)
	≥65	69	(17.0)	98	(24.4)	167	(20.7)
	NA	213	(52.6)	194	(48.3)	407	(50.4)
**Median age (IQR ^b^)**		58.5 (32.2–68.0)		63.0 (52.0–72.0)		61.0 (48.7–70.0)	
**Reporter country**	United States	277	(68.4)	278	(69.2)	555	(68.8)
	Japan	9	(2.2)	70	(17.4)	79	(9.8)
	France	27	(6.7)	-	-	27	(3.3)
	Germany	13	(3.2)	7	(1.7)	20	(2.5)
	Canada	16	(4.0)	2	(0.5)	18	(2.2)
	Switzerland	14	(3.5)	-	-	14	(1.7)
	Israel	-	-	11	(2.7)	11	(1.4)
	Great Britain	8	(2.0)	2	(0.5)	10	(1.2)
	Brazil	10	(2.5)	-	-	10	(1.2)
**Type of reporter**	Consumer	104	(25.7)	185	(46.0)	289	(35.8)
	Physician	138	(34.1)	130	(32.3)	268	(33.2)
	Health professional	110	(27.2)	35	(8.7)	145	(18.0)
	Pharmacist	17	(4.2)	35	(8.7)	52	(6.4)
	Other health professional	35	(8.6)	1	(0.2)	36	(4.5)
	NA	1	(0.2)	16	(4.0)	17	(2.1)
**Reporting year**	2019	78	(19.3)	4	(1.0)	82	(10.2)
	2020	114	(28.1)	62	(15.4	176	(21.8)
	2021	110	(27.2)	136	(33.8	246	(30.5)
	2022	103	(25.4)	200	(49.8)	303	(37.5)

^a.^ ICSR, individual case safety report; ^b.^ IQR, interquartile range.

**Table 3 biomedicines-11-02538-t003:** Distribution of therapeutic indications as reported in larotrectinib and/or entrectinib individual case safety reports (at least accounted for 1.5% of all ICSRs).

Therapeutic Indication	Larotrectinib *N* (%)	Entrectinib *N* (%)	Total *N* (%)
Respiratory tract and pleural neoplasms malignant cell type unspecified NEC ^a^	24 (5.9)	130 (32.3)	154 (19.1)
Non-small cell neoplasms malignant of the respiratory tract cell type specified	11 (2.7)	113 (28.1)	124 (15.4)
Not specified	112 (27.7)	2 (0.5)	114 (14.1)
Therapeutic procedures NEC	3 (0.7)	71 (17.7)	74 (9.2)
Neoplasms malignant site unspecified NEC	23 (5.7)	19 (4.7)	42 (5.2)
Thyroid neoplasms malignant	32 (7.9)	5 (1.2)	37 (4.6)
Breast and nipple neoplasms malignant	20 (4.9)	7 (1.7)	27 (3.3)
Soft tissue sarcomas histology unspecified	21 (5.2)	3 (0.7)	24 (3.0)
Colorectal neoplasms malignant	14 (3.5)	7 (1.7)	21 (2.6)
Neoplasms unspecified malignancy and site unspecified NEC	16 (4.0)	4 (1.0)	20 (2.5)
Pancreatic neoplasms malignant (excluding islet cell and carcinoid)	14 (3.5)	2 (0.5)	16 (2.0)
Glial tumors malignant	11 (2.7)	1 (0.2)	12 (1.5)
Salivary gland neoplasms malignant	12 (3.0)	-	12 (1.5)

^a.^ NEC, not elsewhere classified. NEC is a standard abbreviation used to denote groupings of miscellaneous terms that do not readily fit into other hierarchical classifications within a particular SOC.

**Table 4 biomedicines-11-02538-t004:** Distribution of the preferred term (PT) related to each ADR (*N* = 37) reported in overall individual case safety reports after 365 days.

Preferred Terms	Frequency (*N*)	Frequency (%)
Myalgia	3	8.1%
Fatigue	2	5.4%
Blood thyroid stimulating hormone increased	1	2.7%
Constipation	1	2.7%
Dizziness	1	2.7%
Feeling cold	1	2.7%
Feeling hot	1	2.7%
Flushing	1	2.7%
Hemianesthesia	1	2.7%
Hepatic enzyme increased	1	2.7%
Nausea	1	2.7%
Paresthesia	1	2.7%
Loss of therapeutic response	1	2.7%
Cellulitis	1	2.7%
Hypotension	1	2.7%
Leukopenia	1	2.7%
Sepsis	1	2.7%
Cerebrovascular accident	1	2.7%
General physical health deterioration	1	2.7%
Loss of consciousness	1	2.7%
Nasal aspiration	1	2.7%
Pulmonary thrombosis	1	2.7%
Thrombosis	1	2.7%
Vomiting	1	2.7%
Drug ineffective	1	2.7%
Asthenia	1	2.7%
Eye hemorrhage	1	2.7%
Neuralgia	1	2.7%
Eyelid irritation	1	2.7%
Muscle spasms	1	2.7%
Muscle tightness	1	2.7%
Musculoskeletal stiffness	1	2.7%
Cardiac disorder	1	2.7%
Intracranial mass	1	2.7%

**Table 5 biomedicines-11-02538-t005:** Demographic characteristics of individual case safety reports involving larotrectinib or entrectinib recognized in a spontaneous reporting system, i.e., the FAERS, from 2019–2022 in the pediatric population (≤16 years).

Case n.	Suspect Drug	Sex	Age	Indication	ADR (PT)	TTO ^a^ (Days)	Outcome
**1**	Larotrectinib	NA	10	NA	Vomiting	NA	NA
**2**	Larotrectinib	NA	11	Urinary bladder sarcoma	Renal impairment	NA	Other serious (IME ^b^)
**3**	Larotrectinib	Female	16	Desmoplastic small round cell tumor	Desmoplastic small round cell tumor, Off label use, Product use in unapproved indication, Therapy non-responder	4	Other serious (IME)
**4**	Larotrectinib	Female	12	NA	Rash	NA	NA
**5**	Larotrectinib	Male	14	Congenital fibrosarcoma	Blood alkaline phosphatase increased	NA	NA
**6**	Larotrectinib	NA	1.5	Fibrosarcoma metastatic	Fibrosarcoma metastatic, Metastases to the central nervous system	NA	NA
**7**	Larotrectinib	Male	1	NA	Varicella	NA	Other serious (IME)
**8**	Larotrectinib	Female	14	Soft tissue sarcoma	Vomiting, Intestinal obstruction, Abdominal pain	0	Other serious (IME)
**9**	Larotrectinib	Male	1	Malignant neoplasm of spinal cord	Nephrocalcinosis	NA	Other serious (IME)
**10**	Larotrectinib	Female	16	Sarcoma	Blood disorder, Acute myeloid leukemia, Pancytopenia, Ascites	5	Death
**11**	Larotrectinib	Male	0.5	NA	Lethargy, Somnolence	NA	NA
**12**	Larotrectinib	Female	16	Ovarian melanoma	Multiple organ dysfunction syndrome	NA	Death
**13**	Larotrectinib	NA	0.5	Congenital fibrosarcoma	Body height increased, Weight increased, Growth accelerated	NA	NA
**14**	Larotrectinib	Female	15	Glioneuronal tumor	Gastroenteritis norovirus, COVID-19, Acute kidney injury	NA	Hospitalization
**15**	Larotrectinib	NA	0.5	Congenital fibrosarcoma	Anemia, C-reactive protein increased, Neutrophilia, Leukocytosis	NA	Other serious (IME)
**16**	Larotrectinib	Female	0.5	Soft tissue sarcoma	Cough, Nausea, Pyrexia	NA	NA
**17**	Larotrectinib	NA	1	Neoplasm malignant	Head circumference abnormal	NA	NA
**18**	Entrectinib	Female	16	Neuroendocrine tumor	Disease progression	NA	Other serious (IME)

^a.^ TTO, time to onset; ^b.^ IME, important medical event terms.

## Data Availability

No new data were created to prepare this article, which was entirely based on the review of publicly available ICSRs retrieved from the FAERS.

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
