# Peer review of "The Safety Profiles of Two First-Generation NTRK Inhibitors: Analysis of Individual Case Safety Reports from the FDA Adverse Event Reporting System (FAERS) Database"

_biomedicines, 2023, doi:10.3390/biomedicines11092538_

Round 1
Reviewer 1 Report
Manuscript written by Liguori and colleagues reports safety profile of the first generation NTRK inhibitors. The manuscript is well written and interesting. I would suggest making the minor changes I have listed below.
[1] The Authors write that "Entrectinib was approved by the FDA on August 15, 2019 [...] published by Doebele et al. [16-18]". The references cited as 17-18 are links to CliniclTrials.com. It seems to me that it is better to replace them with references to published articles (if possible).
[2] Please check that there is no error in the notation of the "NTKR" abbreviation used in lines 82, 83 and Table 1.
[3] Line 89: Please explain what ADRs stands for.
[4] Line 102: It is worth briefly explaining what MedDRA is.
[5] Please explain ICSRs used in line 104.
[6] Please explain TTO used in line 175 and Table 1.
Examples of typos:
Line 62: Change “Drilon et colleagues” for “Drilon and colleagues”.
Lines 287 and 231: It seems to me that “SOC” should be changed to “SOCs”.
Manuscript written by Liguori and colleagues reports safety profile of the first generation NTRK inhibitors. The manuscript is well written and interesting. I would suggest making the minor changes I have listed below.
[1] The Authors write that "Entrectinib was approved by the FDA on August 15, 2019 [...] published by Doebele et al. [16-18]". The references cited as 17-18 are links to CliniclTrials.com. It seems to me that it is better to replace them with references to published articles (if possible).
[2] Please check that there is no error in the notation of the "NTKR" abbreviation used in lines 82, 83 and Table 1.
[3] Line 89: Please explain what ADRs stands for.
[4] Line 102: It is worth briefly explaining what MedDRA is.
[5] Please explain ICSRs used in line 104.
[6] Please explain TTO used in line 175 and Table 1.
Examples of typos:
Line 62: Change “Drilon et colleagues” for “Drilon and colleagues”.
Lines 287 and 231: It seems to me that “SOC” should be changed to “SOCs”.
Author Response
Firstly, we would like to thank the reviewer for careful and thorough reading of this manuscript.
Q1: The Authors write that "Entrectinib was approved by the FDA on August 15, 2019 [...] published by Doebele et al. [16-18]". The references cited as 17-18 are links to CliniclTrials.com. It seems to me that it is better to replace them with references to published articles (if possible).
Answer: We have replaced the references to ClinicalTrials.com with references to published articles, as suggested.
Q2: Please check that there is no error in the notation of the "NTKR" abbreviation used in lines 82, 83 and Table 1.
Answer: Indeed, there were errors in the notation of the "NTKR" abbreviation in lines 82, 83 (now lines 90, 91), and Table 1. We have made the necessary corrections.
Q3: Line 89: Please explain what ADRs stands for.
Answer: We have extended the acronym "ADRs" to "Adverse Drug Reactions" to provide a more explicit explanation (now line 97).
Q4: Line 102: It is worth briefly explaining what MedDRA is.
Answer: We have incorporated an explanation of what MedDRA is in the manuscript, as requested (see lines 114-121).
Q5: Please explain ICSRs used in line 104.
Answer: We have extended the acronym “ICSRs” to “individual case safety reports” to provide a more explicit explanation (now line 125).
Q6: Please explain TTO used in line 175 and Table 1.
Answer: We have added a footnote to the Table 5 extending the acronym “TTO” to “time to onset” (line 255).
Q7: Typos
Answers: We have updated from “Drilon et colleagues” to “Drilon and colleagues”. In both instances, "SOC" is used correctly as it refers to a single System Organ Class.
In addition to your suggested revisions, we have also incorporated the following corrections recommended by the other reviewer:
- We provided a figure with the chemical structure of Larotrectinib and entrectinib (Figure 1).
- We have combined the figures 1a. and 1b. into a single one (now Figure 2).
- We have added a period about second-generation NTRK inhibitors in the Introduction (Lines: 81-84).
- We briefly discussed the potential role of concomitant medications on the safety profile of NTRK inhibitors (Lines: 367-371).
- We have corrected the typos.
Reviewer 2 Report
The manuscript summarizes the currently available information about the safety profile of the two clinically approved first-generation NTRK inhibitors larotrectinib and entrectinib. The promising clinical outcome of these NTRK inhibitors warrants a closer look at their side-effects. But several points need to be addressed by the authors in a revised version of the manuscript:
Table 1: Please explain the N in NTRK in the footnotes.
Table 2: Please explain ICSR in the footnotes.
Please provide a figure with the chemical structures of larotrectinib and entrectinib.
Figures 1.a and 1.b should not be separate figures but combined in one figure.
There is no Figure 2a and Figure 3e in the manuscript as stated in the main text, but a Figure 2 and a Figure 2.b instead. Please correct or clarify.
Figure 2: Please explain TTO, time to onset.
Table 5: Please adjust the columns to show the text correctly and properly, not as ´´Cas … e n.´´, ´´Ag … e´´, ´´Femal … e´´ or ´´Larotrectini … b´´.
Maybe the authors can give possible molecular explanations for the shorter TTO of entrectinib when compared with the TTO of larotrectinib.
Please discuss briefly if there is already any information about second-generation NTRK inhibitors in clinical trials, and their adverse effects.
Please discuss how far combinations of larotrectinib and entrectinib with other cancer drugs are under clinical investigation, which might influence adverse effects and safety profile of these NTRK inhibitors.
Line 261: Please correct ´´real-word´´.
n.a.
Author Response
Firstly, we would like to thank the reviewer for careful and thorough reading of this manuscript.
Q1: Table 1: Please explain the N in NTRK in the footnotes.
Answer: We have extended the acronym “NTRK” to “Neurotrophic Tropomyosin Receptor Kinase” in the Table 1 footnotes.
Q2: Table 2: Please explain ICSR in the footnotes.
Answer: We have extended the acronym “ICSR” to “Individual Case Safety Report” in the Table 2 footnotes.
Q3: Please provide a figure with the chemical structures of larotrectinib and entrectinib.
Answer: We provided a figure with the chemical structure of Larotrectinib and entrectinib (Figure 1).
Q4: Figures 1.a and 1.b should not be separate figures but combined in one figure.
Answer: We have combined the figures into a single one (now Figure 2).
Q5: There is no Figure 2a and Figure 3e in the manuscript as stated in the main text, but a Figure 2 and a Figure 2.b instead. Please correct or clarify.
Answer: Indeed, there was an error. We have corrected it (now Figure 3 and Figure 4).
Q6: Figure 2: Please explain TTO, time to onset.
Answer: We have explained TTO in the caption of the Figure 3 (ex Figure 2).
Q7: Table 5: Please adjust the columns to show the text correctly and properly, not as ´´Cas … e n.´´, ´´Ag … e´´, ´´Femal … e´´ or ´´Larotrectini … b´´.
Answer: We have adjusted the columns in table 5 as you suggested.
Q8: Maybe the authors can give possible molecular explanations for the shorter TTO of entrectinib when compared with the TTO of larotrectinib.
Answer: While we acknowledge the interest in exploring possible molecular explanations for the shorter TTO of entrectinib compared to larotrectinib, it's essential to note that the available ICSRs with TTO data were limited in number. Consequently, we prefer not to speculate on potential molecular explanations at this time.
Q9: Please discuss briefly if there is already any information about second-generation NTRK inhibitors in clinical trials, and their adverse effects.
Answer: We have added a period about second-generation NTRK inhibitors in the Introduction (Lines: 81-84).
Q10: Please discuss how far combinations of larotrectinib and entrectinib with other cancer drugs are under clinical investigation, which might influence adverse effects and safety profile of these NTRK inhibitors.
Answer: We briefly discussed the potential role of concomitant medications on the safety profile of NTRK inhibitors (Lines: 367-371).
Q11: Line 261: Please correct ´´real-word´´.
Answer: We have corrected the typo.
In addition to your suggested revisions, We have also incorporated the following corrections recommended by the other reviewer:
- We have replaced the references to ClinicalTrials.com with references to published articles, as suggested (see references 17, 18).
- We have incorporated an explanation of what MedDRA is in the manuscript, as requested (see lines 114-121).
- We have corrected the typos.
Round 2
Reviewer 2 Report
The revised manuscript is suitable for publication now.